# Growth Reduction of *Vibrionaceae* and Microflora Diversity in Ice-Stored Pacific White Shrimp (*Penaeus vannamei*) Treated with a Low-Frequency Electric Field

**DOI:** 10.3390/foods13081143

**Published:** 2024-04-09

**Authors:** Lijuan Xu, Haiqiang Chen, Zuanhao Liang, Shanshan Chen, Yu Xia, Siming Zhu, Ming Yu

**Affiliations:** 1Department of Food and Environmental Engineering, Yangjiang Polytechnic, Yangjiang 529500, China; xulijuan199906@163.com (L.X.); chq888120615@163.com (H.C.); jaly1031@163.com (Z.L.); 2School of Food Science and Engineering, South China University of Technology, Guangzhou 510640, China; 3Guangdong Provincial Engineering and Technology Research Center of Food Low Temperature Processing, Yangjiang 529566, China; 4Institute of Food and Health, Yangtze Delta Region Institute of Tsinghua University Zhejiang, Jiaxing 314006, China; chenshanshan@tsinghua-zj.edu.cn (S.C.); xiayu1984@foxmail.com (Y.X.)

**Keywords:** preservation, shrimp, microbial community, low-frequency electric field, physicochemical properties

## Abstract

A novel storage technique that combines the low-frequency electric field (LFEF) and ice temperature was used to extend the shelf life of Pacific white shrimp (*Penaeus vannamei*). The study investigated the effect of LFEF treatment on the quality and microbial composition of *Penaeus vannamei* during storage at ice temperature. The results showed that the LFEF treatment significantly extended the shelf life of shrimp during storage at ice temperature. The total volatile base nitrogen (TVB-N) and pH of samples increased over time, while the total viable count (TVC) showed a trend of first decreasing and then increasing. Obviously, shrimp samples treated with LFEF had a lower pH, TVB-N and TVC values than the untreated samples (*p* < 0.05) at the middle and late stages of storage. LFEF treatment increased the diversity and altered the composition of the microbial communities in *Penaeus vannamei*. Additionally, the treatment led to a decrease in the relative abundance of dominant spoilage bacteria, including *Aliivibrio*, *Photobacterium* and *Moritella*, in *Penaeus vannamei* stored at ice temperature for 11 days. Furthermore, correlation analysis indicated that TVB-N and pH had a significant and positive correlation with *Pseudoalteromonas*, suggesting that *Pseudoalteromonas* had a greater impact on shrimp quality. This study supports the practical application of accelerated low-frequency electric field-assisted shrimp preservation as an effective means of maintaining shrimp meat quality.

## 1. Introduction

Ice temperature preservation is a technique of storing fresh food at a temperature below 0 °C and above the freezing point to keep it at a low temperature without freezing, typically in the range of −0.5 °C to −2.8 °C [1]. This allows for various biochemical reactions to continue by minimizing cell metabolism. It also works well to maintain cell activity and good tissue structure, inhibiting enzyme activity and bacterial growth. Therefore, ice temperature preservation technology is a safe and efficient method for storing and preserving food, which can improve food quality and extend shelf life to a certain extent [2,3]. This technology has achieved good results in preserving fresh agricultural products, such as fruits, vegetables, livestock, poultry and aquatic products [4]. The application of ice temperature technology in aquatic products is maturing. Previous studies have demonstrated that ice-temperature conditions can enhance the quality of aquatic products, such as extending shelf life, maintaining tissue morphology, and reducing biochemical changes [5]. Although ice temperature preservation methods have been discovered and studied by food scientists since the 1960s, they have not been applied industrially for food storage and preservation [6]. The primary cause of this phenomenon is likely the highly precise temperature control necessary for maintaining optimal ice-temperature conditions. If the temperature fluctuation of the equipment fluctuates outside the required range, it can cause ice crystals to form, ultimately reducing food quality. Additionally, precise temperature control can significantly increase equipment operating costs [4,7], which may limit the use of ice temperature preservation technology in production. Therefore, developing methods to control ice crystal formation could be a crucial breakthrough for the industrial application of ice temperature preservation.

In recent years, researchers have been exploring the use of electric field technology to control the formation of ice crystals [8,9]. Studies have suggested that electric field-assisted supercooling preservation techniques could be effective in maintaining food quality [10,11]. However, the current techniques for controlling ice nucleation through electric fields are not yet fully developed and may not consistently produce positive results [12]. Interestingly, in a preliminary study by our team, it was found that pure water does not freeze at −6 °C under the influence of a low-frequency electric field (LFEF) (frequency range ≤ 300 kHz). Therefore, LFEF may be a potential technology to overcome the technical challenges of industrializing ice temperature preservation.

During the cold temperature storage process, microorganisms are a crucial factor that can affect food quality. Research has confirmed that electric fields can effectively reduce the growth and reproduction of bacteria during subzero storage. Currently, the electrostatic field and pulsed electric field are being studied further. The electrostatic field is a constant field that does not change in strength or direction over time. Static electric fields can cause the rupture of microbial cell structures due to their significant electric field forces. Pulsed electric fields act on food by repeatedly applying a high voltage and short pulses to two parallel metal electrodes to form pulse waves, which involve rapid changes in the electric field and the application of a pulse current [13]. This ultimately destroys the microbial cell structure and acts as a sterilization mechanism. The main purpose of this technique is to sterilize and preserve liquid food. Xu et al. [14] found that applying a high-voltage electrostatic field of different frequencies can assist in preserving the freezing point of pork. This method can effectively inhibit the propagation of dominant bacteria and affect microbial metabolic pathways during storage. Furthermore, Qiang et al. [15] and Zhang et al. [16] investigated the effects of electrostatic field-assisted ice temperatures and partial freezing on the storage quality and microbial communities of rhubarb fish. They found that both electrostatic fields could effectively delay the deterioration of rhubarb fish flesh. However, high-voltage electrostatic fields suppressed the relative abundance of *Shewanella*, while low-pressure fields suppressed the reduction in the relative abundance of *Pseudomonas* and *Psychrobacter*. In contrast to static and pulsed electric fields, low-frequency electric fields, as a type of AC electric field [17], exhibit certain characteristics, such as low electric field intensity (<3 kV) and low frequency (50 and 60 Hz). Low-frequency electric fields, which use space discharge method to form electromagnetic waves and apply them to food, are characterized by simple equipment and system setups, low operational and maintenance costs, and a wide range of applicability, making them suitable not only for large-scale food preservation, but also for use in conjunction with household preservation appliances. Guangyu et al. [18] used alternating electric field technology to assist freeze-thaw aging. They found that alternating the electric field was more conducive to maintaining beef myofibril structure, which promoted meat quality improvement. Kunmei et al. [19] investigated the effect of low-voltage variable-frequency electric fields on the ice temperature preservation effect of steamed mussels, and found that low-voltage variable-frequency electric fields can affect the inventory of microorganisms and better maintain the freshness and quality of mussels. At present, the application of LFEF in the field of food storage and preservation is still at an early stage and deserves further investigation. And studies on shrimp preservation have mainly focused on physical and chemical processes, sensory indicators, volatile metabolites and protein quality changes [20,21,22,23]. However, studies of microbial changes using high-throughput sequencing have been limited, particularly in the context of electric field-assisted cold temperature storage [24,25].

Therefore, in this study, the pH, TVC and TVB-N of *Penaeus vannamei* treated with a low-frequency electric field combined with ice temperature were measured during storage to evaluate quality changes. On the other hand, 16S rRNA amplicon sequencing was performed on fresh shrimp meat from conventional ice temperature storage for 11 days and LEFE combined with ice temperature storage for 11 days to determine the microbiota. The aim was to evaluate the effect of LFEF on the microbial community of shrimp under ice temperature storage conditions and the correlation between colonies and physicochemical indicators, with the aim of elucidating the mechanism of the quality change of *Penaeus vannamei* in ice temperature preservation assisted by LFEF and provide theoretical support for the industrial application of LFEF preservation technology in the field of ice temperature preservation.

## 2. Materials and Methods

### 2.1. Sample Preparation

*Penaeus vannamei* (mariculture) was purchased from the Munai market (Yangjiang, China) and transported alive to the laboratory within 1 h in an oxygenated aquarium. *Penaeus vannamei* was washed with sterile water after sudden death with crushed ice. Shrimps (12–16 g) were pre-cooled at 4 °C for 1 h and randomly divided into two groups: LFEF-assisted ice temperature preservation (A group) and conventional ice temperature preservation (B group). Shrimp were placed in sterile plastic boxes on a clean bench, and then group A samples were stored in a test chamber (−1 ± 1 °C) (LHS-250 CL, Shanghai Qixin Scientific Instrument Co., Ltd., Shanghai, China) with an LFEF generator, and group B samples were stored in a test chamber under the same temperature conditions but without an LFEF generator. Samples were stored at −1 ± 1 °C for 19 days. Shrimp meat samples were taken evenly, where pH and TVB-N were measured every two days and the total viable was measured every three days. Based on the result of significant difference in the TVB-N of ice temperature-stored *Penaeus vannamei* with and without electric field treatment at day 11, we selected fresh *Penaeus vannamei* (XY group), *Penaeus vannamei* (A group) stored in an LFEF combined with ice temperature for 11 days and *Penaeus vannamei* (B group) stored at ice temperature for 11 days for 16S rRNA amplicon sequence analysis. Studies were performed in triplicate.

### 2.2. LFEF Experiment Apparatus

As shown in Figure 1, two transmitting electrodes were placed evenly at the upper and lower ends of the test chamber. The LFEF-generating device outside the test chamber was connected to the transmitting electrode inside the tube by a transmission wire. The LFEF-generating device was assembled according to the authorized patent of our team. The device consisted of an input control module, an electric field signal generation module, an output control module, an electric field generation module, and a safety protection module.

### 2.3. Determination of pH during Storage

Shrimp meat sample (5 g) and distilled water (50 mL) were homogenized at 9000 rpm/min for 1 min and the pH was measured using a digital pH meter (Yidian Scientific Instrument Ltd., Shanghai, China) [26].

### 2.4. Determination of TVC during Storage

Shrimp meat (5 g) was placed in a sterile bag containing saline (45 mL). The mixture was homogenized for 2 min using a homogenizer and the suspension of homogenized samples was serially diluted (1:10). The sample solution (l mL) at the appropriate dilution was added to 20 mL of a plate count agar medium, solidified, and then incubated at (36 ± 1 °C) for (48 ± 2) h. The results of the experiments were expressed as log (CFU/g) [27].

### 2.5. Determination of TVB-N during Storage

The stranded sample (5 g) was mixed with 50 mL of distilled water in a conical flask. The mixture was shaken and macerated for 30 min, filtered to obtain the filtrate. Of the filtrate, 5 mL and a MgO suspension (10 g/L) were accurately aspirated into the reaction chamber and distilled with Kjeldahl (Hongji Instrument Ltd., ATN-300, Shanghai, China) for 5 min. The receiving bottle was used with a boric acid solution as the absorbent and then titrated with 0.01 mol/L standard hydrochloric acid solution to the end point. The TVB-N value was calculated from the hydrochloric acid consumption. The results are expressed as mg/100 g of shrimp meat [28].

### 2.6. DNA Extraction

Bacterial genomic DNA was extracted from the samples using the CTAB method [29]. Shrimp muscles were added to liquid nitrogen, and the appropriate amount was transferred to sterilized EP tubes after repeated grinding, and total DNA was extracted using the Soil and Fecal Genomic DNA Extraction Kit by Magnetic Bead Method (DP712, Tiangen Biochemistry & Technology Co., Ltd., Beijing, China). The DNA concentration and purity were monitored on a 1% agarose gel. After extraction, the DNA was dissolved in 50 μL of elution buffer and stored at −80 °C until it was sent to Novogene Bio-information Science and Technology Co., Ltd. (Beijing, China) for polymerase chain reaction amplification and sequencing.

### 2.7. PCR Amplification and Illumina Sequencing

Depending on the concentration, the DNA was diluted to 1 ng/µL with sterile water. PCR amplification was performed using the diluted DNA as the template and 515F-806R (5′ GTGCCAGCMGCCGCGGTAA-3′ and 5′-GGACTACHVGGGTWTCTAAT-3′) as primers. The amplification was performed in a 30 μL reaction volume containing 10 ng of the genomic DNA extract, 15 μL Phusion^®^ High-Fidelity PCR Master Mix (New England Biolabs, Ipswich, MA, USA) and 3 μL of each primer, and the mixture was made up to 30 μL with water. The PCR conditions were as follows: initial denaturation for 1 min at 98 °C and 30 cycles of amplification (10 s at 98 °C, 30 s at 50 °C, 30 s at 72 °C). The last step was a final extension for 5 min at 72 °C. According to the concentration of the PCR products, the samples were mixed in equal concentrations, the mixture of PCR products was purified using the Qiagen Gel Extraction Kit (Qiagen, Hilden, Germany), and the sequences with the major bands between 400 and 450 bp were selected and trimmed to recover the target bands. Tag-encoded high-throughput sequencing was performed on the Illumina HiSeq platform and 250 bp paired-end reads were generated.

### 2.8. Processing of the Sequencing Data

The original data were spliced (FLASH: fast length adjustment of short reads to improve the genome assemblies) and filtered (chimeric 16S rRNA sequence formation and detection in Sanger and 454-pyrosequenced PCR amplicons) to obtain the clean data, and then the operational taxonomic unit (OTU) clustering and species classification were analyzed based on the clean data. The SILVA database (The SILVA ribosomal RNA gene database project: improved data processing and web-based tools) was used for each representative sequence, based on the Mothur algorithm to annotate the taxonomic information. Alpha diversity was used to analyze the complexity of species diversity for a sample through 6 indexes, including Observed-species, Chao1, Shannon, Simpson, ACE, and Good coverage. All these indexes in our samples were calculated using QIIME (version 1.7.0). Based on the results of species annotation, clustering heat-map and Linear discriminant analysis [30] were plotted using R software (version 2.15.3). Finally, the relationship between bacterial flora and physicochemical factors was preliminarily explained by correlation analysis.

### 2.9. Statistical Analysis

Data were analyzed by an analysis of variance (ANOVA) using SPSS 17.0 software. Duncan’s test was used to determine the significance of differences in the experimental data within the 95% confidence interval (*p* < 0.05). Differences between the two sample datasets were determined by independent sample *t*-tests. Origin Lab Origin 2022b software was used to plot the graphs in the experiment.

## 3. Results and Discussion

### 3.1. pH Value in Shrimp during Storage

As shown in Figure 2a, the pH value of shrimp meat in both groups increased significantly with storage time (*p* < 0.05). The pH value increased relatively quickly during the early storage period, but relatively slowly during the later storage period. This was caused by the decomposition of shrimp protein under the action of shrimp microorganisms and endogenous enzymes after shrimp death, resulting in the accumulation of nitrogenous substances. During the early storage period, there was no significant difference in the pH between the two groups. From the 11th day, the pH value of group A was significantly lower than that of group B. This could be due to the fact that the LFEF affected the degradation of organic compounds by microorganisms in shrimp meat during the later stages of storage, thus delaying the accumulation of final products. This is similar to the results of low-frequency pulsed electric field treatment of refrigerated Pacific white shrimp by Shiekh et al. [22].

### 3.2. TVC in Shrimp during Storage

The total viable count reflects the changes in the microbial population during the storage of the shrimp. As shown in Figure 2b, the initial total colony count was 3.74 lg (CFU/g). With increasing storage time, both groups showed a trend of a slight decrease followed by a sharp increase. On the 19th day of storage, the TVC of group B exceeded 6 lg (CFU/g), indicating that the shrimp meat was unacceptable [31]. From the 9th day of storage, there was a significant difference between the two groups (*p* < 0.05). And the TVC of shrimp meat in group B increased faster. This is consistent with the results of Jianyou Zhang’s study [16] on a large yellow croaker, where the total number of colonies was lower in the electric field-treated group than in the control group. In shrimp at the same storage temperature, the electric field treatment had some inhibitory effect on bacterial growth and multiplication compared to group B. This may be due to the fact that under the action of the applied electric field, the electric potential inside and outside of the microbial cell membrane was altered, resulting in a decrease in microbial viability [32].

### 3.3. TVB-N Value in Shrimp during Storage

TVB-N is one of the biochemical indicators that effectively indicate shrimp spoilage. As shown in Figure 2c, the TVB-N content of both preservation methods showed an obvious increasing trend with storage time. The TVB-N value of shrimp samples treated with group B reached 28.37 mg/100 g on the 11th day, which was close to the borderline between inedible and spoiled seafood (≤30 mg/100 g). However, the shrimp samples in group A did not deteriorate and became spoiled up to the 19th day, and the TVB-N value increased to 33.59 mg/100 g. This shows that the influence of an LFEF on the ice temperature preservation of shrimp meat significantly impacts the TVB-N value (*p* < 0.05). LFEF can effectively inhibit the spoilage of shrimp, which may be due to the inhibitory effect of the electric field on the growth of microorganisms. This would affect the metabolism of spoilage microorganisms and reduce the production of nitrogenous substances. This is consistent with the conclusion that LFEF is favorable for reducing the total number of colonies, as concluded in Section 3.3 of this study.

### 3.4. Sequencing Data Analysis

Paired-end sequence data were obtained using the Illumina Nova sequencing platform. By splicing the reads, an average of 97,472 tags were measured per sample. The Chao1 and ACE indexes (Table 1) were higher than the number of OTUs obtained in all the sample groups, indicating that almost all the bacteria in the samples were detected, and some other bacterial phylotypes were present [33]. The Goods coverage of all sample groups was greater than 0.9, indicating that the sequencing information data were credible and could be used to analyze the microbial diversity of the samples.

Sequences were clustered into operational taxonomic units (OTUs) with 97% identity, and a total of 2332 OTUs were obtained. The observed species, Chao 1, and Ace indexes were used to assess the abundance of microbial communities. The Shannon and Simpson indexes reflect the diversity of the community and the concentration of dominant species, respectively [34]. As shown in Table 1, compared to the XY group, the Observed species, ACE, and Chao1 indexes decreased, while the Shannon and Simpson indexes increased in the B group. In general, species richness decreases during spoilage as the dominant spoilage organisms dominate the colony [35], and species diversity increases during the later stages of storage as the role of specific spoilage organisms decreases [36]. A higher Simpson index indicates a more prominent dominant bacterium in the community. Group B samples had the highest Simpson index, which could be attributed to the formation of a dominant flora. Interestingly, group A showed the highest indexes in this study, which were very different from some previous studies [14,28,37,38], indicating that the shrimp samples in group A had the richest and most diverse bacterial species. The bactericidal effect of LFEF may have inhibited the growth and reproduction of some dominant bacteria, leading to an increase in microbial diversity in the LFEF treatment group. This could be due to the production of bacteria that constantly compete for limited substrates and nutrients [39]. Moreover, bacteria have a mechanism of mutual regulation among themselves. The metabolites secreted by individual microorganisms are acquired by others, enabling some individuals to survive in the community environment, forming a pattern of interdependence.

### 3.5. Species Annotation and Composition Analysis

Figure 3 reveals changes in bacterial composition and abundance within *Penaeus vannamei* samples under different treatment conditions. The microbial composition of aquatic products undergoes significant changes during the early stages of storage, typically resulting in a decrease in microbial abundance and diversity. As storage time increases, only few bacteria become the major spoilage bacteria during the late stages of spoilage [40,41]. Figure 3a shows that at the phylum level, *Proteobacteria* and *Firmicutes* dominated each group (XY, A, B), which is consistent with previous studies [41,42]. *Proteobacteria* was the dominant phylum in all three sample groups, but was more abundant in group B. In addition, the relative abundance of *Firmicutes* decreased in group B compared to group XY, while there were no significant changes in the abundance of *Proteobacteria* and *Firmicutes* in the LFEF-treated samples. *Proteobacteria* are commonly associated with the spoilage of aquatic products, with most of the microorganisms responsible belonging to this phylum [43,44]. The abundant presence of *Proteobacteria* in group B samples may be attributed to the nutrient-rich nature of spoiled shrimp, which facilitates their growth. As anticipated, the bacterial community profile of LFEF-treated shrimp was similar to that of fresh shrimp. This may be attributed to the ability of LFEF to inhibit *Proteobacteria* and *Firmicutes* from growing during storage.

At the family level (Figure 3b), the initial dominant bacteria in fresh shrimp samples were *Burkholderiaceae*, *Vibrionaceae* and *Moraxellaceae*, accounting for 51.93%, 15.32% and 3.31%, respectively. After 11 days of storage at ice temperature, there was a significant increase in the percentage of *Vibrionaceae*, *Moritellaceae* and *Pseudoalteromonadaceae* in shrimp muscle, accounting for 28.09%, 8.29% and 7.74%, respectively. Burkholderiaceae are widespread in various environments, including pathogenic bacteria [45]. The presence of a large number of *Burkholderiaceae* in fresh shrimp was mainly due to the growth environment. The *Vibrionaceae* family, which contains a variety of bacteria that are commonly found as specific spoilage bacteria in aquatic products, is mainly distributed in marine environments [46]. Under ice temperature storage conditions of −1 ± 1 °C, *Vibrionaceae* was the absolute spoilage bacterium causing shrimp spoilage. Interestingly, the relative abundance of *Vibrionaceae* in the shrimp muscle of group A was significantly lower, while *Pseudoalteromonadaceae* and *Moritellaceae* tended to become dominant. This indicates that the LFEF had a good inactivation effect on *Vibrionaceae*. Mittenzwey et al. [47] observed changes in the luminescence of the marine luminescent bacteria *Photobacterium phosphoreum* and *Photobacterium fisheri* in the presence of electromagnetic fields ranging from 2 to 50 Hz. The study found that exposure to electromagnetic fields of 50 Hz and 2 mT suppressed the luminescence of the bacteria. However, exposure to electromagnetic fields of other frequencies and intensities did not have an impact on the luminescence of the bacteria. This study observed a significant decrease in the abundance of *Vibrionaceae* following low-frequency electric field treatment. It is hypothesized that certain bacteria within *Vibrionaceae* may be more sensitive to low-frequency electric fields, which affects their metabolic processes and leads to growth and reproduction inhibition.

At the genus level (Figure 3c), the bacterial species initially found in shrimp samples were *Ralstonia*, *Vibrio* and *Aliivibrio*. These genera are known to originate from the intestinal tract of *Penaeus vannamei*, according to previous studies [48]. In group B, the dominant bacteria were *Ralstonia*, *Alivibrio*, *Photobacterium*, *Moritella* and *Pseudoalteromonas*. In group A, the most prevalent genera were *Pseudoalteromonas* and *Psychrobacter*, suggesting that LFEF may influence the bacterial community structure by regulating the relative abundance of the dominant genera.

*Ralstonia* is a genus of Gram-negative bacteria that is commonly found in soil and water [49]. There have been reports that this bacterium may be a potential route for the transmission of foodborne bacteria [50], but it is inherently less infectious. It is important to control the safety risks posed by this bacterium due to its high abundance throughout the storage period. After 11 days of storage under different storage conditions, the relative abundance of dominant bacteria in the samples changed significantly. The Vibrionaceae family includes the Gram-negative genera *Aliivibrio* and *Photobacterium*, which are closely related [51]. These genera are known to be strong contributors to fish and shrimp spoilage and can produce large amounts of biogenic amines during storage, leading to off-flavors [42,52]. The odor of the group B samples in this study quickly became unacceptable, possibly due to the abundance of them at the end of the storage period. *Moritella* is a genus that has been influenced as a contributor to the spoilage of marine aquatic products during storage [51,53]. In this study, its abundance increased toward the end of the storage period. *Moritella* is a bacterium that thieves in cold, high-pressure environments [54]. The electric field can directly or indirectly affect microorganisms by altering their cellular structure and affecting the permeability of microbial cells [55], which in turn affects their metabolism [56] and activity, ultimately inhibiting their growth. In group A, the relative abundance of *Alivibrio*, *Photobacterium* and *Moritella* significantly decreased after LFEF treatment. The results indicate that LFEF treatment can effectively slow down spoilage by reducing the abundance of these spoilage bacteria.

In addition, LFEF was ineffective in inhibiting the growth of *Pseudoalteromonas* and *Psychrobacter*. *Pseudoalteromonas*, the primary genus responsible for shrimp spoilage, produces substantial amounts of sulfides that hydrolyze proteins [57,58]. Iijima et al. [59] demonstrated that *Pseudoalteromonas* is capable of forming robust biofilms. The bacterial cells within the biofilm are typically surrounded by extracellular polymeric substances, making them highly resistant to physicochemical stresses. *Psychrobacter* species are typically considered moderate spoilers, as they are unable to compete with common spoilage microorganisms [28,57]. However, *Psychrobacter* promoted the growth of *Pseudoalteromonas* and there are interactions between them that enhance spoilage activity [60]. Therefore, when applying LFEF preservation, it is important to control both *Pseudoalteromonas* and *Psychrobacter*.

### 3.6. Heatmaps of Species Composition and Key Species Differences

A heatmap was generated to analyze and compare the relative abundances and changes in the microbial community at the genus level (top 35 genera) in all samples (Figure 4). Clustering analysis revealed significant genus-level differences in microbiota among all three groups. Higher microbial diversity was observed in groups XY and A, whereas in group B, microbial diversity showed a decreasing trend. However, the relative abundance of spoilage bacteria remained at a high level and increased significantly, which is consistent with the previous result of the α-diversity analysis.

Linear discriminant analysis (LDA) effect size (LEfSe) classification was used to identify biomarkers that were statistically different between the groups [30] (Figure 5). A threshold of an LDA score > 4 (*p* < 0.05) was used for differential screening, and a two-way comparison was performed between groups A and B. The results showed that several characteristics of the 11-day samples treated with different storage conditions exhibited highly significant differences from order to genus. Group B shrimp samples were enriched with *Vibrionales*, *Vibrionaceae*, *Photobacterium* and *Photobacterium phosphoreum*. These bacteria belong to the same evolutionary branch, which further confirms the inhibitory effect of LFEF on *Vibrionaceae*.

### 3.7. Correlation Analysis

The correlation between physicochemical factors and microorganisms is crucial for preserving shrimp meat. TVB-N and pH are important indications of deterioration in aquatic products. Figure 6 displays the results of the Spearman correlation analysis which was used to analyze the correlation between TVB-N and pH values with the top 10 abundant microbial compositions at the genus level as inputs. Notably, the TVB-N and pH as indicators of deterioration showed a significant positive correlation. *Pseudoalteromonas*, *Aliivibrio*, *Moritella*, *Shewanella* and *Photobacterium* were found to have a positive correlation with TVB-N and pH. Among them, *Pseudoalteromonas* showed a significant positive correlation. These bacteria are believed to play a major role in the spoilage process of aquatic products due to their high production of TVB-N [60,61]. In addition, Figure 6 demonstrates a significant positive correlation between these bacteria, suggesting a synergistic relationship that may promote mutual growth [62,63]. Understanding the relationship between the main dominant strains is crucial for improving preservation methods for shrimp meat [40], as microbial interactions play a significant role in determining its composition.

## 4. Conclusions

This study evaluated the effects of a low-frequency electric field on pH, TVC, TVB-N and microbial community of *Penaeus vannamei* at ice temperature during storage. The results showed that low-frequency electric field treatment inhibited bacterial growth, while reducing the increase in pH and TVB-N values. The results of high-throughput sequencing showed that LFEF treatment prolonged the shelf life of shrimp muscle by increasing the diversity of microbial communities and reducing the abundance of spoilage bacteria to maintain the stability of the colony structure. Both treatment groups experienced significant changes in shrimp flora abundance after 11 days of storage. The relative abundance of *Vibrionaceae*, which was beneficial in delaying shrimp meat spoilage, was significantly reduced in LFEF-treated samples. However, *Pseudoalteromonas* and *Psychrobacter* showed tolerance to the low-frequency electric fields when combined with ice temperature environments. Correlation analysis revealed a positive correlation between the TVB-N and pH with *Pseudoalteromonas*, and there were also correlations among the major genera. Therefore, it is important to control the growth of *Pseudoalteromonas* when applying low-frequency electric fields for the low temperature preservation of shrimp in the future.

## Figures and Tables

**Figure 1 foods-13-01143-f001:**
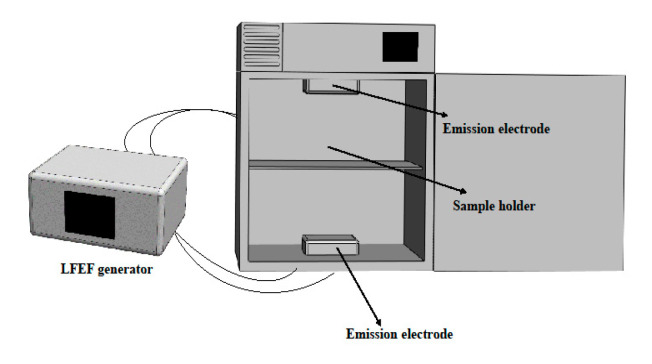
Schematic diagram of the low-frequency electric field-assisted ice temperature preservation equipment used in this study.

**Figure 2 foods-13-01143-f002:**
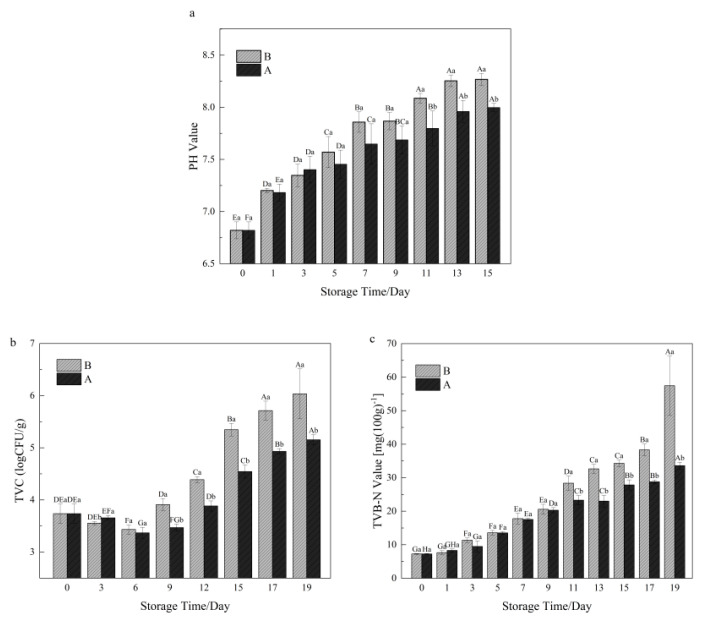
Changes in pH (**a**), TVC (**b**) and TVB-N (**c**) during the storage of *Penaeus vannamei*. A: LFEF-assisted storage of shrimp at ice temperature; B: shrimp stored at ice temperature. Different capital letters indicate significant differences within the same treatment group (*p* < 0.05). Different lower-case letters indicate significant differences between groups for the same storage time (*p* < 0.05); analysis was performed in triplicate.

**Figure 3 foods-13-01143-f003:**
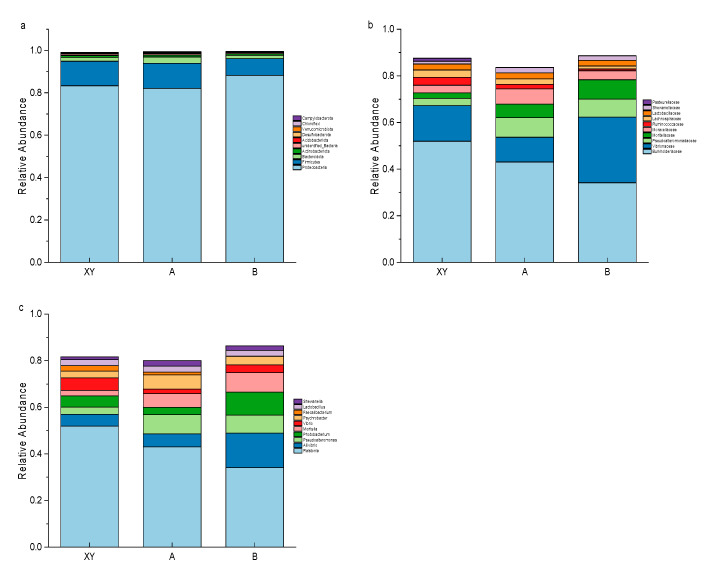
Relative abundance of bacteria in the three groups of shrimp samples at the phylum level (**a**), family level (**b**) and genus levels (**c**). (XY: fresh shrimp; A: LFEF-assisted ice temperature storage of shrimp for 11 days; B: shrimp stored at ice temperature for 11 days).

**Figure 4 foods-13-01143-f004:**
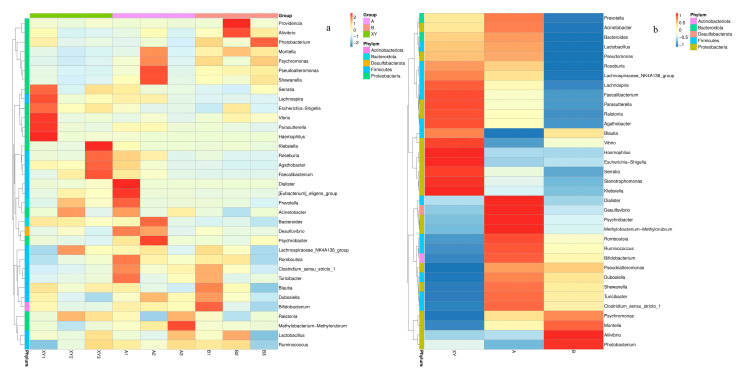
Heatmap of bacteria at the top 35 genus levels in all shrimp samples (**a**) and the three groups (**b**). The red squares indicate higher concentrations of the substances, while the blue squares showcase lower concentrations. (XY1–XY3: the sample collected from the fresh shrimp; A1–A3: the sample collected from the LFEF-assisted ice temperature storage of shrimp for 11 days; B1–B3: the sample collected from the shrimp stored at ice temperature for 11 days.).

**Figure 5 foods-13-01143-f005:**
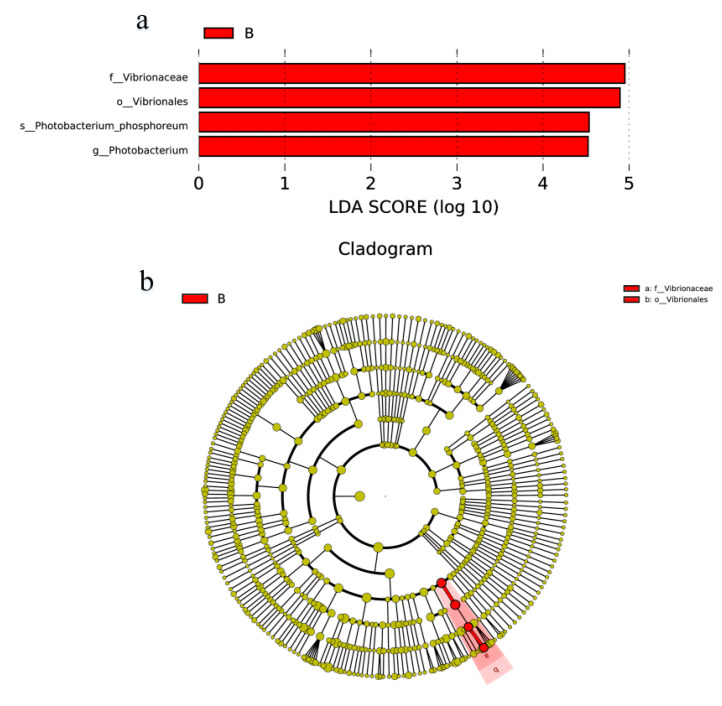
Bacterial genera discriminating between samples with LFEF treatment from those untreated, calculated by linear discriminant analysis (LEfSe). LDA scores (logarithmic LDA score ≥ 4.0 and *p* ≤ 0.05) (**a**) and cladogram (**b**). In the cladogram, the circles from the inside to the outside represent the classification levels from phylum to species. Coloring principle: species with no significant differences are colored yellow, and the biomarker of different species is colored following the group. The red nodes represent the microbial groups that play an important role in the red group. If a certain group is missing in the figure, it means that there are no significantly different species in this group regarding taxa, so this group is missing.

**Figure 6 foods-13-01143-f006:**
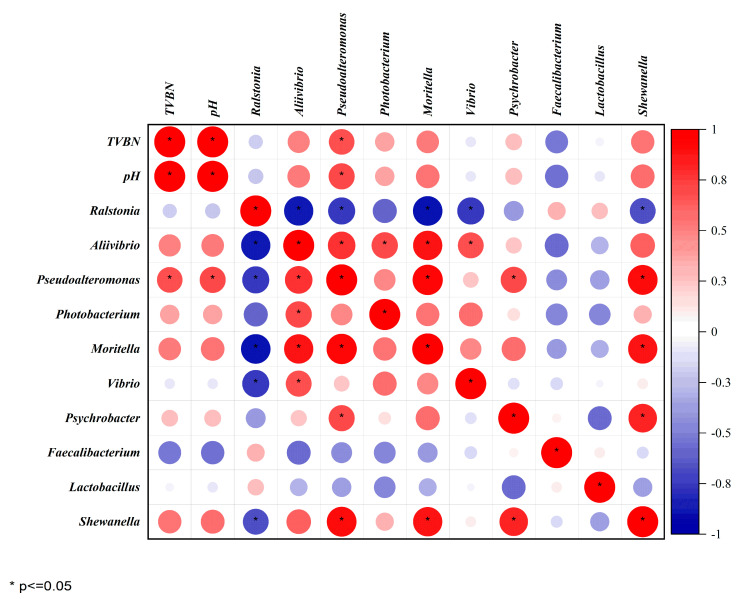
Heatmap of the Spearman correlation between the TVB-N, pH and the top 10 genera in shrimps. The degree of color represents the absolute value of the correlation coefficient, where red represents a positive correlation and blue represents a negative correlation. (*. Significantly correlated at the 0.05 level (bilateral)).

**Table 1 foods-13-01143-t001:** Alpha diversity profiles of the three groups of samples of shrimps, (values are the mean ± standard deviation of three determinations).

Alpha Diversity	ACE	Chao1	Shannon	Simpson	Observed Species	Goods Coverage	PD_Whole_Tree
XY	1162.905 ± 48.073	1113.229 ± 41.160	4.003 ± 0.590	0.709 ± 0.1	984.000 ± 48.073	0.995 ± 0.001	125.648 ± 13.310
B	1034.974 ± 88.8	997.573 ± 84.427	4.243 ± 0.421	0.835 ± 0.0	838.333 ± 88.861	0.995 ± 0.001	83.566 ± 10.884
A	1207.065 ± 66.651	1162.186 ± 73.308	4.390 ± 0.999	0.765 ± 0.172	1005.667 ± 66.651	0.995 ± 0.000	96.578 ± 13.408

## Data Availability

The original contributions presented in the study are included in the article, further inquiries can be directed to the corresponding authors.

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
