# Peer review of "Growth Reduction of Vibrionaceae and Microflora Diversity in Ice-Stored Pacific White Shrimp (Penaeus vannamei) Treated with a Low-Frequency Electric Field"

_foods, 2024, doi:10.3390/foods13081143_

Round 1

Reviewer 1 Report

Comments and Suggestions for Authors

Review

Title: Growth reduction of Vibrionaceae and microflora diversity in 2 ice-stored Pacific white shrimp (Penaeus vannamei) treated 3 with low-frequency electric field

Abstract: Does not display top search results

key words: keywords must be designed to be more efficient, avoiding repetition of words in the title

Introduction

Line 34:  and its temperature range is typically -0.5 to 2.81”  There is an authorship error. This information about the initial freezing point interval of most foodstuffs was provided by Fennema, Powrie, & Marth, 1973, cited in the Introduction section of the article by Duun and Rustad.

Line 78: Corret to: In addition, Qiang et al. 15 and Zhang et al.,16 investigated the …

Material and methods:

Lines 126/127: total viable count every three days. Commente: Total viable count of what? Bacterial cells?

Line 161: 2.6. DNA extractionSequencing sample preparation????.

Results and discussion

Lines 197-199: …relationship between flora and physicochemical factors…Comment: microbial flora? Or bacterial flora??

Lines 209-211: redundant sentence

Lines 216-217: he reference used to compare the results of pH variation in shrimp tissues between the treatments applied does not present data on this variable in its study.

Lines 309-312: “Interestingly, the relative abundance of Vibrionaceae in the shrimp muscle of group A was significantly lower, while Pseudoalteromona- daceae and Moritellaceae tended to become dominant, indicating that the LFEF had a good inactivation effect on Vibrionaceae”. Comment: What is the explanation for this effect on bacterial cells of the genus Vibrio???

Lines 323-324: “Ralstonia and Vibrio in the initial flora are pathogenic bacteria and generally do not have specific spoilage effects”. Comment:  The statement is not correct and the citation from Lacerda et al., 2022 is inappropriate.

Lines 334/335: “The spoilage potential of Moritella in shrimp spoilage has not been reported, but Moritella is closely related to the genus Shewanella 42. Comment: Authors must review articles used as references. This statement does not belong to the article cited. There are articles listing species of the genus Moratella as spoilage of marine aquatic foods. Proximity to Shewanella refers to one species ( M. marina previously Vibrio marinus ) and not the entire genus.

Comments on the Quality of English Language

some sentences are unclear and repetitive

Reviewer 2 Report

Comments and Suggestions for Authors

The main purpose of this study was to assess the efficacy of a novel storage technique utilizing a combination of low-frequency electric field (LFEF) and ice temperature to extend the shelf life of Pacific white shrimp (Penaeus vannamei). The research investigated the impact of LFEF treatment on the quality and microbial composition of the shrimp during storage at ice temperature. The results demonstrated that LFEF treatment better maintained key quality indicators such as pH, total viable count (TVC), and Total Volatile Base Nitrogen (TVB-N), ultimately prolonging the shelf life of the shrimp by at least 6 days. The idea is good but there are numerous points that should be considered before publication:

- The similarity index, 67%, is very high. Please reduce this.

- Please don’t reuse the words that you already used in title for keywords.

- Please give some sentences about the motivation of this work in the abstract.

- How many replicates did you do. This information is missing.

- Statistical analysis: Please give detail in this part. What was the decision criteria for significant differences. Please rewrite this part.

- Figure 2 needs to be revised. First visual quality should be improved. Second, what are capital and small letters referring? Please put explanation bottom up the figure.

-Figure 3: legends could not be read. The same thing is for fig. 4 and fig. 5.

-Conclusions: Please don’t use active voice sentence. Recheck all throughout the article.

Comments on the Quality of English Language

Moderate editing of English language required.

Round 2

Reviewer 2 Report

Comments and Suggestions for Authors

From my side, it is ready for publication.

Comments on the Quality of English Language

Minor editing of English language required.

Author Response

Thank you for your approval of our article.